# R2-Dreamer: Redundancy-Reduced World Models without Decoders or Augmentation

**Naoki Morihira**[1,2]     **Amal Nahar**[1]     **Kartik Bharadwaj**[1]     **Yasuhiro Kato**[2]
**Akinobu Hayashi**[1,2]     **Tatsuya Harada**[2,3]
[1] Honda R&D Co., Ltd.     [2] The University of Tokyo     [3] RIKEN AIP

## ABSTRACT

A central challenge in image-based Model-Based Reinforcement Learning (MBRL) is to learn representations that distill essential information from irrelevant visual details. While promising, reconstruction-based methods often waste capacity on large task-irrelevant regions. Decoder-free methods instead learn robust representations by leveraging Data Augmentation (DA), but reliance on such external regularizers limits versatility. We propose R2-Dreamer, a decoder-free MBRL framework with a self-supervised objective that serves as an internal regularizer, preventing representation collapse without resorting to DA. The core of our method is a *redundancy-reduction* objective inspired by Barlow Twins, which can be easily integrated into existing frameworks. On DeepMind Control Suite and Meta-World, R2-Dreamer is competitive with strong baselines such as DreamerV3 and TD-MPC2 while training $1.59\times$ faster than DreamerV3, and yields substantial gains on DMC-Subtle with tiny task-relevant objects. These results suggest that an effective internal regularizer can enable versatile, high-performance decoder-free MBRL. Code is available at `https://github.com/NM512/r2dreamer`.

## 1 INTRODUCTION

Learning effective latent representations is a cornerstone of world models in Model-Based Reinforcement Learning (MBRL), yet this poses a significant challenge: representations must capture task-essential information without overfitting to irrelevant details. While architectures like the Recurrent State-Space Model (RSSM) have achieved remarkable success (Hafner et al., 2025), a fundamental question remains open: **What is the optimal objective function for learning the representation itself?** This question is particularly important in image-based settings, where high-dimensional observations make representation learning inherently challenging.

In practice, many leading methods learn representations by optimizing pixel-level reconstruction objectives (Micheli et al., 2023; Zhang et al., 2023; Seo et al., 2023; Micheli et al., 2024; Alonso et al., 2024; Hafner et al., 2025). This creates a critical issue: the learning signal is dominated by spatially large but task-irrelevant parts of the observation, such as the background. Consequently, the model is incentivized to meticulously reconstruct these details, wasting representational capacity and computational resources at the expense of ignoring small but task-critical objects.

To address the limitations of pixel-wise reconstruction, decoder-free methods learn representations via self-supervised losses (Deng et al., 2022; Okada & Taniguchi, 2022; Burchi & Timofte, 2025). To prevent the representation collapse common in such approaches, they depend critically on Data Augmentation (DA) as an external regularizer. This reliance on DA is a significant bottleneck for general agents (Laskin et al., 2020; Ma et al., 2025), as the choice of transformation is task-dependent: random shifting can discard crucial small objects, while color jittering can be detrimental when color itself is a key feature.

In this work, we focus on the representation learning objective within the widely used RSSM framework and propose **R2-Dreamer** that breaks the dependency on decoders and DA. To isolate the impact of the learning objective itself, we build upon the well-established Dreamer architecture. Inspired by Barlow Twins (Zbontar et al., 2021), we introduce a *redundancy-reduction* objective between image embeddings and latent states to prevent representation collapse without external regularizers, providing a versatile and robust baseline capable of achieving competitive performance.

Our main contributions are:

- A new representation learning paradigm for RSSM-based decoder-free MBRL that replaces heuristic DA, which risks distorting task-critical information, with an internal redundancy reduction objective.
- Competitive performance across standard benchmarks, including DeepMind Control Suite (DMC) and Meta-World, and superior performance on our new, challenging DMC-Subtle benchmark, while enabling faster training by removing the decoder.
- The release of our unified PyTorch codebase, including implementations of our method and baselines built on our DreamerV3 implementation, along with the DMC-Subtle benchmark to facilitate future research.

## 2 RELATED WORK

Our work is positioned at the intersection of MBRL and Self-Supervised Learning (SSL). We contextualize our approach by reviewing representation learning strategies in MBRL and how they address the challenge of regularization.

### 2.1 REPRESENTATION LEARNING IN WORLD MODELS

**Decoder-Based World Models**   A dominant paradigm in MBRL, popularized by the Dreamer series (Hafner et al., 2025), learns representations by reconstructing observations from a latent state. While successful, this reconstruction-based objective often forces the model to waste capacity on task-irrelevant details, such as backgrounds, motivating a shift towards decoder-free methods.

**Decoder-Free World Models and the Reliance on DA**   To address the limitations of reconstruction, recent decoder-free methods learn representations through auxiliary objectives that do not involve pixel-wise reconstruction, such as predicting future rewards or learning via contrastive losses. However, despite the diversity in their learning signals, these prominent examples (Ye et al., 2021; Deng et al., 2022; Hansen et al., 2022; 2024; Wang et al., 2024; Burchi & Timofte, 2025) all critically rely on DA—typically random shifts—as an external regularizer to prevent representation collapse. This fundamental dependency on augmentations that may distort task-relevant details limits their versatility, a key bottleneck we address.

Aside from DA, some methods mitigate visual distractions through architectural mechanisms; for instance, VAI (Wang et al., 2021) introduces additional attention modules but relies on motion cues, which can overlook static yet task-critical visual cues. Several works regularize representations more directly by injecting Gaussian noise into latent features (Shu et al., 2020; Nguyen et al., 2021). In contrast, we show that a single information-theoretic principle for redundancy reduction is sufficient for stable and effective representation learning in RSSM-based models without any DA.

### 2.2 FROM INVARIANCE TO INFORMATION-BASED REGULARIZATION

**DA-Driven Invariance**   Many popular self-supervised representation learning methods, including those used in existing decoder-free agents, are invariance-based. They rely on DA to create positive pairs (e.g., augmented views of the same image) and train the model to produce similar representations for them, as seen in contrastive (Chen et al., 2020; He et al., 2020; Caron et al., 2020) and non-contrastive (Grill et al., 2020; Chen & He, 2021) learning. In this paradigm, DA is essential to prevent collapse to trivial solutions.

**DA-Free Internal Regularization**   Our work adopts a different approach from the information-based SSL literature (Zbontar et al., 2021; Bardes et al., 2022), which focuses on reducing feature redundancy. While these methods still use DA in the computer vision domain, we adapt this principle to serve as a complete replacement for DA-based regularization in the reinforcement learning domain. Specifically, we apply the redundancy reduction objective between the image encoder's output and the RSSM's latent state. This yields an *internal regularizer* sufficient to prevent representation collapse, thereby allowing us to build a more versatile and robust learning framework without task-specific augmentations.

## 3 METHOD

Our method, R2-Dreamer, redesigns the representation learning mechanism of the powerful DreamerV3 (Hafner et al., 2025) framework to be decoder-free and DA-free. We achieve this by replacing its reconstruction-based objective with a self-supervised objective based on redundancy reduction, inspired by Barlow Twins (Zbontar et al., 2021). To isolate the impact of our proposed learning objective, other components of the world model and the actor-critic implementation are kept identical to the original DreamerV3. This single change demonstrates notable improvements in computational efficiency and robustness. This section first details the latent dynamics model, introduces our new world model learning objective, and reviews the actor-critic learning process.

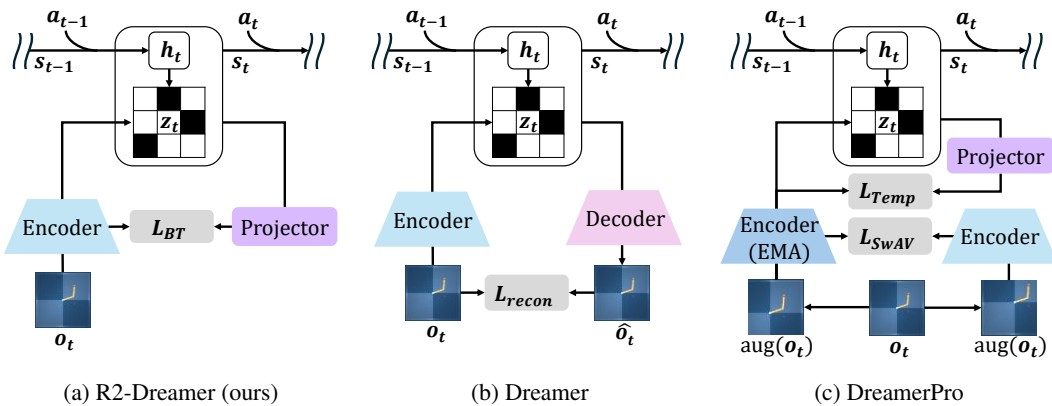

| (a) R2-Dreamer (ours) | (b) Dreamer | (c) DreamerPro |
| :---: | :---: | :---: |

Figure 1: Comparison of representation learning mechanisms in world models. **(a) R2-Dreamer** learns representations without a decoder or DA. It uses an internal redundancy reduction objective $\mathcal{L}_{\text{BT}}$ that aligns the latent state $s_t$ (via a projector) with the embedding of the observation $o_t$. **(b) Dreamer** relies on a decoder to learn representations by reconstructing the observation $\hat{o}_t$ from the latent state $s_t$, guided by a reconstruction loss $\mathcal{L}_{\text{recon}}$. **(c) DreamerPro** removes the decoder but depends on DA. It enforces consistency between augmented views of the observation $\text{aug}(o_t)$ using a spatial loss $\mathcal{L}_{\text{SwAV}}$ and a temporal loss $\mathcal{L}_{\text{Temp}}$ that leverages an Exponential Moving Average (EMA) of the encoder weights.

### 3.1 LATENT DYNAMICS MODEL

Following DreamerV3, we use the RSSM (Hafner et al., 2019) with a composite latent state $s_t = (h_t, z_t)$, consisting of a deterministic state $h_t$ and a stochastic state $z_t$. This state acts as the agent's memory, aggregating the sequence of observations $o_t$ and actions $a_t$ to model dynamics, and it is also used to predict rewards $r_t$ and continuation flags $c_t$.

The key architectural distinction of R2-Dreamer is the **complete removal of the image decoder** and the introduction of a lightweight linear projector head. While DreamerV3 relies on a decoder to reconstruct observations $\hat{o}_t \sim p_\phi(\hat{o}_t \mid s_t)$ for representation learning, our projector simply maps the latent state $s_t$ to the feature space of the image embedding $e_t$. This design avoids the computational cost of pixel-level reconstruction while enabling effective representation learning through the objective described in the next section. The components of our model are defined as follows:

$$
\begin{aligned}
\textbf{Image Encoder:} \quad & e_t = f_\phi(o_t) \\
\textbf{Sequence Model:} \quad & h_t = f_\phi(s_{t-1}, a_{t-1}) \\
\textbf{Dynamics Predictor:} \quad & \hat{z}_t \sim p_\phi(\hat{z}_t \mid h_t) \\
\textbf{Representation Model:} \quad & z_t \sim q_\phi(z_t \mid h_t, e_t) \\
\textbf{Reward Predictor:} \quad & \hat{r}_t \sim p_\phi(\hat{r}_t \mid s_t) \\
\textbf{Continue Predictor:} \quad & \hat{c}_t \sim p_\phi(\hat{c}_t \mid s_t) \\
\textbf{Projector:} \quad & k_t = f_\phi(s_t)
\end{aligned}
\tag{1}
$$

## 3.2 World Model Learning

Our core contribution is a new learning objective for the world model that replaces the reconstruction loss of DreamerV3. As theoretically motivated in Appendix A, this new objective is a tractable surrogate for an extended Sequential Information Bottleneck objective. We now detail the practical implementation of this objective, adhering to the original loss components from DreamerV3 where applicable.

**DreamerV3 Objective**  The world model in DreamerV3 is trained by optimizing four distinct objectives: reconstruction, prediction, and two KL-divergence terms for regularizing the latent dynamics. The overall loss, shown in Eq. equation 2, is a weighted sum of these components.

$$\mathcal{L}_{\text{DreamerV3}}(\phi) = \mathbb{E}_{q_\phi} \left[ \sum_t \left( \mathcal{L}_{\text{recon}}(t) + \mathcal{L}_{\text{pred}}(t) + \beta_{\text{dyn}} \mathcal{L}_{\text{dyn}}(t) + \beta_{\text{rep}} \mathcal{L}_{\text{rep}}(t) \right) \right] \qquad (2)$$

The reconstruction and prediction losses are negative log-likelihoods. The dynamics and representation losses are regularized using KL balancing (Hafner et al., 2021) and free bits (Kingma et al., 2016). Each component is defined as:

$$\begin{aligned}
\mathcal{L}_{\text{recon}}(t) &= -\log p_\phi(o_t|s_t) \\
\mathcal{L}_{\text{pred}}(t) &= -\log p_\phi(r_t|s_t) - \log p_\phi(c_t|s_t) \\
\mathcal{L}_{\text{dyn}}(t) &= \max\left(1, \text{KL}\big[\text{sg}(q_\phi(z_t|h_t, e_t)) \,\|\, p_\phi(z_t|h_t)\big]\right) \\
\mathcal{L}_{\text{rep}}(t) &= \max\left(1, \text{KL}\big[q_\phi(z_t|h_t, e_t) \,\|\, \text{sg}(p_\phi(z_t|h_t))\big]\right)
\end{aligned} \qquad (3)$$

where sg denotes the stop-gradient operator.

**R2-Dreamer Objective**  We remove the reconstruction term $\mathcal{L}_{\text{recon}}$ and replace it with our proposed loss, $\mathcal{L}_{\text{BT}}$. The other components, including the KL balancing scheme and loss coefficients ($\beta_{\text{dyn}} = 1$, $\beta_{\text{rep}} = 0.1$), are adopted from DreamerV3:

$$\mathcal{L}_{\text{world}}(\phi) = \mathbb{E}_{q_\phi} \left[ \beta_{\text{BT}} \mathcal{L}_{\text{BT}} + \sum_t \left( \mathcal{L}_{\text{pred}}(t) + \beta_{\text{dyn}} \mathcal{L}_{\text{dyn}}(t) + \beta_{\text{rep}} \mathcal{L}_{\text{rep}}(t) \right) \right] \qquad (4)$$

This formulation isolates the contribution of our method to the new representation learning signal provided by $\mathcal{L}_{\text{BT}}$.

**Representation Learning via Redundancy Reduction ($\mathcal{L}_{\text{BT}}$)**  We adopt the Barlow Twins objective as our redundancy reduction mechanism. Compared to other methods like VICReg (Bardes et al., 2022), it is chosen for its minimal implementation footprint and fewer hyperparameters, which reduces tuning effort. The objective is defined as:

$$\mathcal{L}_{\text{BT}} = \underbrace{\sum_i \left(1 - \mathbf{C}_{ii}\right)^2}_{\text{Invariance Term}} + \alpha \underbrace{\sum_{i \neq j} \mathbf{C}_{ij}^2}_{\text{Redundancy Term}} \qquad (5)$$

The indices $i$ and $j$ refer to the feature dimensions. Here, $\mathbf{C}$ is the cross-correlation matrix computed over a mini-batch between the projector output $k_t$ and the image embedding $e_t$ (see Appendix G for details). This loss is governed by a single hyperparameter, $\alpha$, which weights the redundancy reduction term. Instead of creating artificial views via DA, we form a natural pair of views from the model's internal signals: the image embedding $e_t$ and the projected latent state $k_t$.

In practice, we compute $\mathbf{C}$ over all time steps in a mini-batch (i.e., over $B \times T$ samples) after standardizing both $k_t$ and $e_t$ along the batch dimension.

In our implementation, we detach the target $e_t$ to enhance stability, similar to strategies in TD-MPC2 (Hansen et al., 2024). Despite this, the encoder receives rich gradients backpropagating through the projector and RSSM, while the reward, episode continuation, dynamics, and value objectives provide task-relevant supervision identical to DreamerV3.

### 3.3 ACTOR-CRITIC LEARNING

To ensure our performance gains are attributable to the world model's representation quality, the actor-critic learning process remains unchanged from DreamerV3. The critic is optimized on both imagined rollouts and replay trajectories, whereas the actor is optimized only on imagined trajectories. Specifically, imagined rollouts start from latent states inferred from replayed trajectories and are unrolled using the learned dynamics model under the current policy.

The critic is trained to predict the distribution of $\lambda$-returns, a robust estimate of future rewards. The critic's loss is the maximum likelihood of predicting these returns:

$$\mathcal{L}_{\text{critic}}(\psi) = -\mathbb{E}_{p_\phi, \pi_\theta}\left[\sum_{t=1}^{H} \log p_\psi(R_t^\lambda | s_t)\right] \tag{6}$$

where the $\lambda$-return $R_t^\lambda$ is computed recursively as $R_t^\lambda = r_t + \gamma c_t\big((1-\lambda)v_\psi(s_{t+1}) + \lambda R_{t+1}^\lambda\big)$, with discount $\gamma$ and continuation flag $c_t$. As in the original setup, this objective is applied to both imagined rollouts and replay trajectories from the buffer.

The actor is trained to maximize these returns using the REINFORCE estimator (Williams, 1992), incorporating entropy regularization with a fixed scale $\eta$ and robust return normalization:

$$\mathcal{L}_{\text{actor}}(\theta) = -\mathbb{E}_{p_\phi, \pi_\theta}\left[\sum_{t=1}^{H}\left(\text{sg}\left(\frac{R_t^\lambda - v_\psi(s_t)}{\max(1, S)}\right)\log \pi_\theta(a_t|s_t) + \eta\text{H}[\pi_\theta(a_t|s_t)]\right)\right] \tag{7}$$

where $S$ is a dynamically scaled normalizer computed as an exponential moving average of the 5–95$^{\text{th}}$ percentile range of returns, i.e., $S \doteq \text{EMA}(\text{Per}(R_t^\lambda, 95) - \text{Per}(R_t^\lambda, 5), 0.99)$, which improves robustness to outliers across diverse environments.

## 4 EXPERIMENTS

In this section, we conduct a series of experiments to validate the core claims of our work: that R2-Dreamer learns high-quality representations in a decoder-free and DA-free manner, leading to a framework that is not only computationally efficient but also highly performant. Our evaluation is structured to answer the following key questions:

1. How does R2-Dreamer perform against leading decoder-based and decoder-free agents on standard continuous control benchmarks? (Sec. 4.2, Sec. 4.3)

2. How does our internal regularization handle challenging scenarios where task-relevant information is subtle and easily missed by competing methods? (Sec. 4.4)

3. How does the learned representation qualitatively differ from baselines in focusing on task-relevant information? (Sec. 4.5)

4. What is the direct impact of our proposed redundancy reduction objective compared to other design choices, particularly DA? (Sec. 4.6)

5. What are the computational benefits of its decoder-free and DA-free design in practice? (Sec. 4.7)

We report task scores on DMC and DMC-Subtle and success rates on Meta-World, summarizing results with mean and median across tasks, and provide detailed per-task curves in the appendix. In all experiments, we conduct training over five random seeds, with 10 evaluation episodes per seed, and, unless otherwise stated, use the same hyperparameter configuration (see Appendix F) across all tasks and benchmark suites.

### 4.1 EXPERIMENTAL SETUP

**Baselines**  We compare R2-Dreamer against a carefully selected set of competitive baselines to cover the main paradigms of image-based reinforcement learning:

- **R2-Dreamer (ours)**: Implemented on top of our PyTorch-based DreamerV3 reproduction. This unified codebase is used for all decoder-free variants to ensure that performance differences are directly attributable to the representation learning objective.

- **DreamerV3** (Hafner et al., 2025): A leading and highly competitive decoder-based world model. To provide one of the strongest and most credible baselines, we use the author's official JAX implementation as our primary point of comparison, utilizing the latest version which includes several algorithmic improvements made in April 2024.[1]

- **Dreamer-InfoNCE**: A contrastive learning baseline using the InfoNCE loss (van den Oord et al., 2019) to investigate performance in the absence of DA, implemented on our DreamerV3 reproduction.

- **DreamerPro** (Deng et al., 2022): A leading decoder-free method that relies on DA, specifically random image shifts, to prevent representation collapse. Since the original implementation is based on DreamerV2, we re-implemented its core mechanism on our DreamerV3 reproduction to ensure a fair comparison. This re-implementation also improved its performance.

- **DrQ-v2** (Yarats et al., 2021): A strong and widely-used model-free agent for image-based RL, included as a representative model-free baseline for performance reference. It relies on DA as a key component of the method. We use the author's official implementation.[2]

- **TD-MPC2** (Hansen et al., 2024): A strong decoder-free model-based method that combines TD learning with planning in latent space, and uses DA as an external regularizer to prevent representation collapse. We use the author's official implementation.[3]

**Environments** All our benchmarks focus on continuous control from pixels. We evaluate our method on three benchmark suites:

- **DeepMind Control Suite (DMC)** (Tassa et al., 2018): A widely adopted benchmark suite for continuous control tasks from pixels, encompassing both locomotion and manipulation domains.

- **Meta-World** (Yu et al., 2021): A benchmark suite for evaluating performance on diverse manipulation tasks using a robotic arm. We use the MT1 benchmark, where agents are trained on 50 distinct tasks individually. These tasks involve interacting with various objects, including small ones, and require precise fine-grained manipulation.

- **DMC-Subtle**: A new benchmark designed as a controlled stress test for representation learning in pixel-based control, where task-critical objects are scaled down to make task-relevant visual cues subtle. For example, Figure 2 illustrates the Reacher task, where the target is scaled down to one-third of its original size. This benchmark demands a higher level of representational precision. Detailed modifications for all tasks are provided in Appendix B.

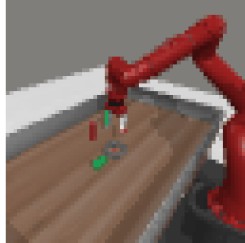 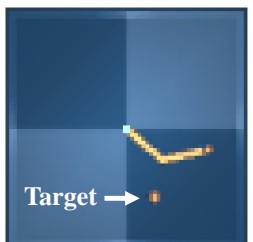 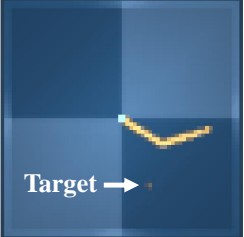

Figure 2: Examples drawn from benchmark tasks. Left: Meta-World Assemble. Center: DMC Reacher (hard). Right: DMC-Subtle Reacher with a significantly smaller target.

---

[1] https://github.com/danijar/dreamerv3
[2] https://github.com/facebookresearch/drqv2
[3] https://github.com/nicklashansen/tdmpc2

## 4.2 PERFORMANCE ON DEEPMIND CONTROL SUITE

We first evaluate R2-Dreamer on 20 standard DMC tasks. Figure 3 summarizes performance across tasks using both the mean and the median. Our method is competitive with the decoder-based, decoder-free, and model-free baselines on average. This result indicates that our internal redundancy reduction objective is an effective learning signal, capable of achieving competitive performance without a decoder or an external regularizer like DA. Detailed per-task curves are in Appendix C.

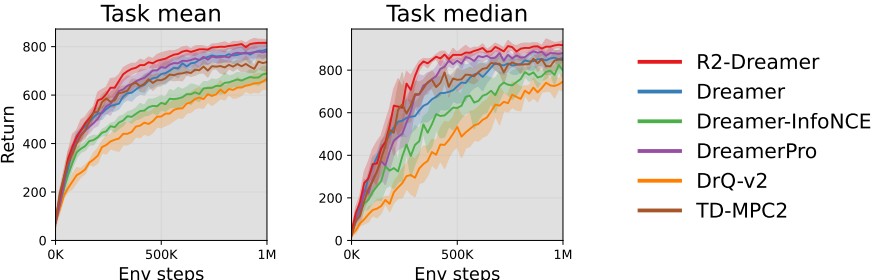

Figure 3: Mean and median performance over 20 DMC tasks, with standard deviation across seeds. R2-Dreamer is competitive with the baselines on average without requiring a decoder or DA.

## 4.3 PERFORMANCE ON META-WORLD

We evaluate R2-Dreamer on Meta-World MT1, which contains 50 robotic manipulation tasks trained independently. Figure 4 reports the mean and median success rate across tasks, with standard deviation across seeds. R2-Dreamer is competitive with the baselines in mean success rate across tasks on average, even on contact-rich manipulation tasks involving small objects. Detailed per-task curves are in Appendix D.

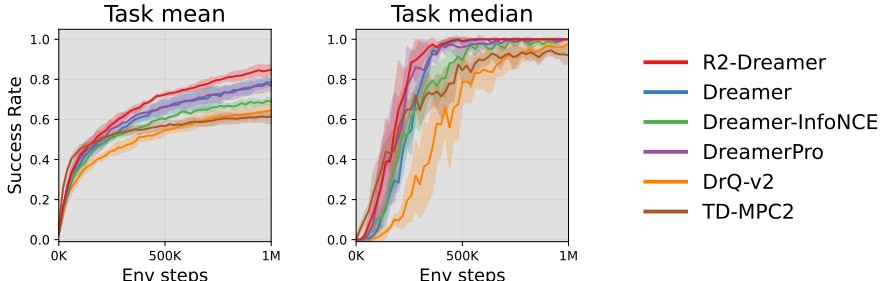

Figure 4: Aggregated performance on Meta-World 50 tasks using the mean and the median across tasks. R2-Dreamer achieves strong results even on contact-rich manipulation tasks, remaining competitive with the baselines on average.

## 4.4 ROBUSTNESS IN CHALLENGING ENVIRONMENTS

We now highlight the benefits of our approach on the DMC-Subtle benchmark, a challenging testbed designed to penalize methods that either overfit to irrelevant backgrounds or discard small, critical objects. We hypothesize that our redundancy reduction objective is particularly well-suited for these precision-demanding tasks. By not being driven by a reconstruction signal dominated by task-irrelevant backgrounds and avoiding the potential distortion of critical features from DA, our method should learn more focused representations. The results in Figure 5 confirm this hypothesis, showing a substantial performance gap over the baselines and demonstrating that R2-Dreamer can effectively isolate and attend to task-critical information, a crucial capability for real-world applications where salient cues may be sparse. We further analyze the learned representations to understand the source of this robustness.

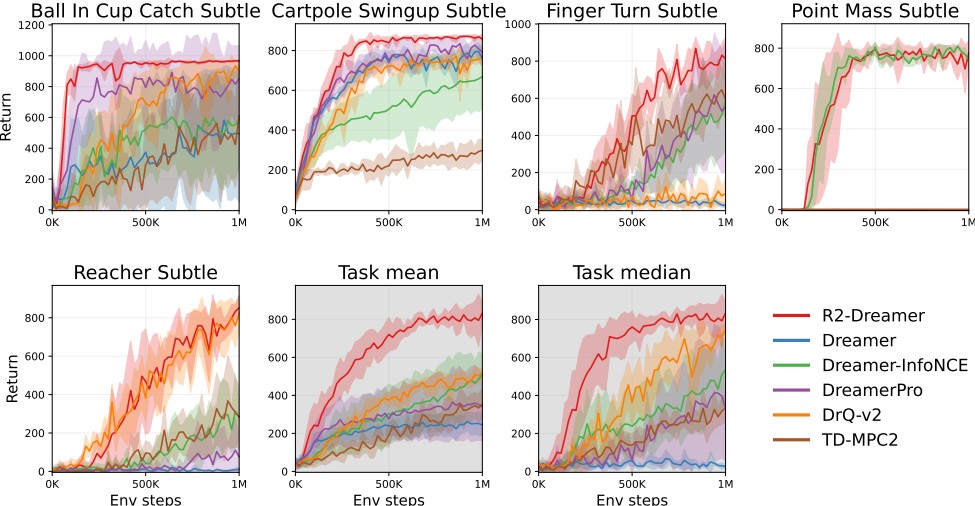

Figure 5: Performance on five challenging DMC-Subtle tasks. R2-Dreamer substantially outperforms the baselines, demonstrating its robustness to subtle but critical visual information.

## 4.5 ANALYSIS OF LATENT REPRESENTATIONS

We visualize the policy's focus using an occlusion-based saliency method (Greydanus et al., 2018) to assess how well the learned representations capture task-relevant information. For this analysis on the DMC-Subtle Reacher task, we compute saliency maps on the first frame of each episode to isolate the spatial focus from temporal dynamics. The results in Figure 6 reveal a clear distinction: the saliency map for R2-Dreamer is sharply focused on the target, indicating its policy is grounded in task-critical visual evidence. In contrast, baselines exhibit more diffuse saliency, suggesting a less precise understanding of the task. This finding provides strong qualitative evidence that our redundancy reduction objective encourages learning compact and relevant representations.

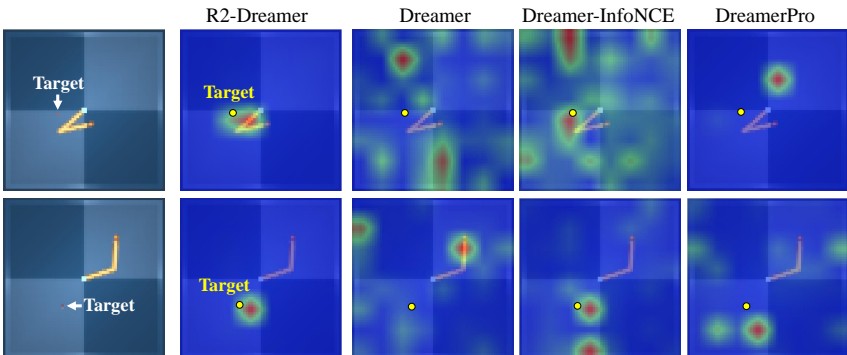

Figure 6: Policy saliency maps on the DMC-Subtle Reacher task. For clarity, the target location is marked with a yellow dot. The two rows show results from two different environment seeds, which are identical across all methods.

## 4.6 ABLATION STUDIES

To isolate our core contributions, we conduct a targeted ablation study on the effectiveness of our redundancy reduction objective against DA. We compare six variants: **R2-Dreamer** (our complete method), **R2-Dreamer (half batch)**, **R2-Dreamer with DA** (adding random shift), **DreamerPro** (DA-reliant baseline), **DreamerPro without DA**, and **Dreamer without Decoder** (no visual auxiliary objective).

First, Figure 7 shows that adding DA to R2-Dreamer yields only marginal gains. In contrast, DreamerPro collapses without DA, confirming its critical dependency on the external regularizer. Performance degrades close to that of Dreamer without Decoder, which has no explicit objective to learn visual representations.

We also test batch-size sensitivity, as SSL objectives can be affected by correlation estimation. Consistent with the reported robustness of Barlow Twins (Zbontar et al., 2021), halving the batch size ($B = 8$ vs. $B = 16$) does not yield a significant performance drop.

Detailed results are in Appendix E.

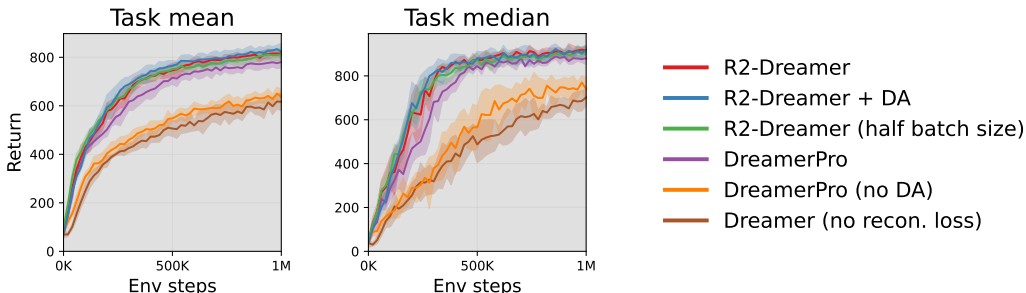

Figure 7: Ablation results on 20 DMC tasks using the mean and the median across tasks. Our internal redundancy reduction objective proves more effective and robust than reliance on heuristic DA.

Second, we examine the same design choice in a setting where preserving fine-grained spatial information is essential. On the precision-demanding DMC-Subtle benchmark, DA proves detrimental. As shown in Figure 8, adding DA significantly degrades our method's performance. This highlights a key risk of external regularizers: while generally applicable, they can distort subtle, task-critical information. Our DA-free, internal mechanism provides a more robust solution in such cases, reinforcing its effectiveness as a principled regularizer for RSSM.

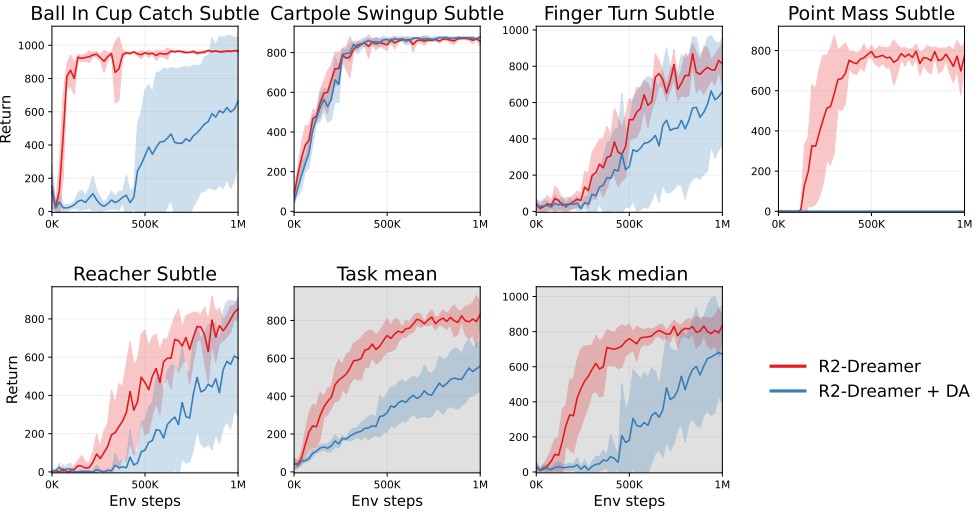

Figure 8: Comparison of R2-Dreamer with and without DA on the DMC-Subtle benchmark. The results highlight that DA can be detrimental in tasks requiring high precision, as it may distort subtle but critical information, underscoring the importance of a DA-free approach for such environments.

### 4.7 COMPUTATIONAL EFFICIENCY

A core advantage of our decoder-free design is its computational efficiency. To ensure a fair comparison, we measure the wall-clock training time of our method against baselines implemented on our unified DreamerV3 reproduction. As shown in Table 1, R2-Dreamer achieves a $1.59\times$ speedup over our DreamerV3 reproduction by eliminating the computationally expensive image generation process. Furthermore, it demonstrates a $2.36\times$ speedup compared to DreamerPro, which involves processing different augmented views of the input and subsequent relatively complex logic. We also include the training time of the original, highly optimized DreamerV3 JAX implementation. These results highlight that R2-Dreamer offers a more practical and scalable solution.

Table 1: Computational efficiency comparison on the DMC Walker Walk task. Wall-clock time is measured for 1 million environment steps on a single NVIDIA GeForce RTX 3080 Ti GPU.

| Method | Training Time (hours) |
| --- | --- |
| R2-Dreamer | **4.4** |
| Dreamer | 7.0 |
| DreamerPro | 10.4 |
| Dreamer (Author's JAX impl.) | 6.6 |

## 5 CONCLUSION

We demonstrated that a principled internal regularization objective can supplant the need for image reconstruction in MBRL. Our framework, R2-Dreamer, learns representations focused on salient features without decoders or task-specific DA.

The strength of this approach is most evident on our challenging DMC-Subtle benchmark, where R2-Dreamer substantially outperforms leading decoder-based and DA-reliant agents by isolating tiny, critical objects. On standard benchmarks across locomotion and manipulation domains, it is competitive with DreamerV3 while achieving $1.59\times$ faster training.

An important direction for future work is to evaluate R2-Dreamer in environments with dynamic and irrelevant backgrounds, such as the Distracting Control Suite (Stone et al., 2021). Our results on DMC-Subtle suggest that our internal redundancy reduction objective naturally avoids wasting representational capacity on irrelevant pixels, which may imply robustness to such dynamic distractors. Verifying this hypothesis would further establish the efficacy of DA-free internal regularization for complex visual control tasks. Scaling to high-dimensional tasks like Humanoid is also a future direction.

By shifting the focus from visual fidelity to informational efficiency, our work provides a scalable foundation for building agents where heuristic augmentations risk distorting task-critical information. This study opens a new inquiry into internal regularization as a principled path toward more general and capable learning agents.

### REPRODUCIBILITY STATEMENT

To ensure reproducibility, we provide detailed methodology (Sec. 3), experimental setups (Sec. 4.1), and hyperparameters (Appendix F). Pseudocode is available in Appendix G, and we release the unified PyTorch codebase including R2-Dreamer, baselines built on our DreamerV3 implementation, and the DMC-Subtle benchmark.

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

# A  Connecting Redundancy Reduction to the Sequential Information Bottleneck

Our World Model's loss function optimizes a variational bound on an extended Sequential Information Bottleneck (SIB) objective (Tishby et al., 2000). Building on DreamerV1 (Hafner et al., 2020), our formulation incorporates a spatial compression term that encourages disentanglement by minimizing the Total Correlation (TC) (Watanabe, 1960) of the latent states. The full objective is defined as:

$$\max \quad \underbrace{\mathrm{I}\big(s_{1:T}; (o_{1:T}, r_{1:T}, c_{1:T}) \mid a_{1:T}\big)}_{\text{Fidelity}} - \underbrace{\beta\,\mathrm{I}\big(s_{1:T}; i_{1:T} \mid a_{1:T}\big)}_{\text{Temporal Compression}} - \underbrace{\gamma \sum_{t=1}^{T} \mathrm{TC}(s_t)}_{\text{Spatial Compression}} \tag{8}$$

where $s_{1:T}$ is the latent state sequence, $(o_{1:T}, r_{1:T}, c_{1:T})$ are the observation, reward, and continuation sequences, and $a_{1:T}$ is the action sequence. Following (Alemi et al., 2017), $i_t$ denotes the underlying data-generating index for the observation. The objective of compressing $s_{1:T}$ with respect to $i_{1:T}$ is to discard predictable information from the past, thereby encouraging the latent state to capture only novel information. Below, we derive tractable variational bounds for each term and demonstrate how our proposed loss function optimizes them.

**Lower Bound on Fidelity**  The fidelity term ensures that the latent state $s_{1:T}$ retains predictive information about observation, reward, and continuation. As this term is intractable, we maximize a variational lower bound. The derivation begins with the chain rule for mutual information. For simplicity, considering only observations $o_{1:T}$:

$$\begin{aligned} \mathrm{I}(s_{1:T}; o_{1:T} \mid a_{1:T}) &= \sum_{t=1}^{T} \mathrm{I}(s_{1:T}; o_t \mid o_{1:t-1}, a_{1:T}) \\ &\geq \sum_{t=1}^{T} \mathrm{I}(s_t; o_t \mid o_{1:t-1}, a_{1:T}) \\ &\geq \sum_{t=1}^{T} \mathrm{I}(s_t; e_t \mid o_{1:t-1}, a_{1:T}) \\ &\approx \sum_{t=1}^{T} \mathrm{I}(s_t; e_t) \end{aligned} \tag{9}$$

The first inequality holds as information cannot increase with a subset of variables, and the second follows from the data processing inequality ($e_t = \mathrm{enc}(o_t)$). The final step drops the conditioning on history under the common assumption that $s_t$ is a sufficient statistic of the past (Hafner et al., 2019), under which the approximation preserves the inequality direction. A similar derivation, omitting the data processing inequality step, can be applied to rewards and continuation signals. This yields the final surrogate objective for the fidelity term:

$$\mathrm{I}\big(s_{1:T}; (o_{1:T}, r_{1:T}, c_{1:T}) \mid a_{1:T}\big) \gtrsim \sum_{t=1}^{T} \mathrm{I}\big(s_t; e_t\big) + \sum_{t=1}^{T} \mathrm{I}\big(s_t; r_t\big) + \sum_{t=1}^{T} \mathrm{I}\big(s_t; c_t\big) \tag{10}$$

**Upper Bound on Temporal Compression**  Following prior work (Hafner et al., 2020), the temporal compression term is upper-bounded by the KL divergence between the posterior and prior dynamics:

$$\mathrm{I}\big(s_{1:T}; i_{1:T} \mid a_{1:T}\big) \leq \sum_{t=1}^{T} \mathbb{E}_q\Big[D_{\mathrm{KL}}\big(q(s_t|s_{t-1}, a_{t-1}, o_t) \,\big\|\, p(s_t|s_{t-1}, a_{t-1})\big)\Big] \tag{11}$$

**Unification via Barlow Twins**  Crucially, the extended SIB objective's fidelity and spatial compression terms can be jointly optimized by a single surrogate loss based on the Barlow Twins objective (Eq. equation 5). This loss is applied to the image embedding $e_t$ and the projected state $k_t$, and consists of two components:

- **Invariance**: The loss penalizes the deviation of the diagonal elements of the cross-correlation matrix from 1. This encourages the projected state $k_t$ to predict the image embedding $e_t$, a surrogate for maximizing the fidelity term $\mathrm{I}(s_t; e_t)$.

- **Redundancy Reduction**: The loss penalizes the off-diagonal elements of the cross-correlation matrix. This encourages the dimensions of $k_t$ to be uncorrelated. Since $k_t$ is a linear projection of the latent state $s_t = (h_t, z_t)$, i.e., $k_t = W[h_t; z_t]$, this pressure to decorrelate $k_t$ directly incentivizes the model to learn a factorized representation. This, in turn, aligns the optimization to minimize the Total Correlation, thus approximating the spatial compression objective.

This unified objective provides a practical and theoretically motivated mechanism for representation learning. While the SIB framework (Eq. equation 8) uses theoretical coefficients $\beta$ and $\gamma$, our practical loss function (Eq. equation 4) implements these compression terms as a collection of weighted surrogate losses, including the KL balancing (Hafner et al., 2021).

# B    DETAILED DESCRIPTIONS OF DMC-SUBTLE TASKS

This section details all five tasks in the DMC-Subtle benchmark introduced in Section 4.3. Figure 9 compares the standard version of each task with our modified version, where task-critical objects have been intentionally scaled down to just a few pixels. The specific modifications are as follows:

- **Ball in Cup Catch**: The agent must swing a tethered ball into a cup. The ball size and string width are reduced to 1/12 of the original.

- **Cartpole Swingup**: The agent must swing up and balance a pole on a cart. The pole width is reduced to 1/20 of the original.

- **Finger Turn**: The agent must spin a two-link finger to touch a target. The target size is reduced to 1/2 of the original.

- **Point Mass**: The agent must move a point mass to a target. The goal is removed as it is always at the center, and the point mass size is reduced to 1/6 of the original.

- **Reacher**: The agent must guide a two-link arm to reach a target. The target size is reduced to 1/3 of the original.

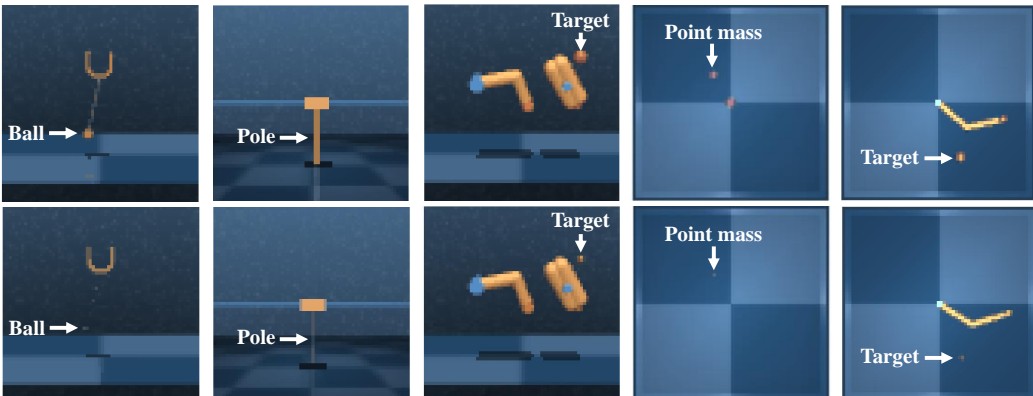

Figure 9: DMC-Subtle benchmark. Top: original versions of the five tasks (left to right: Ball in Cup Catch, Cartpole Swingup, Finger Turn, Point Mass, Reacher). Bottom: modified versions with downscaled task-critical objects in the same order.

# C    DETAILED RESULTS ON DEEPMIND CONTROL SUITE

This section provides the individual learning curves for all 20 tasks in the DMC benchmark (Figure 10). The corresponding aggregated results (mean/median across tasks) are shown in Figure 3 in the main text.

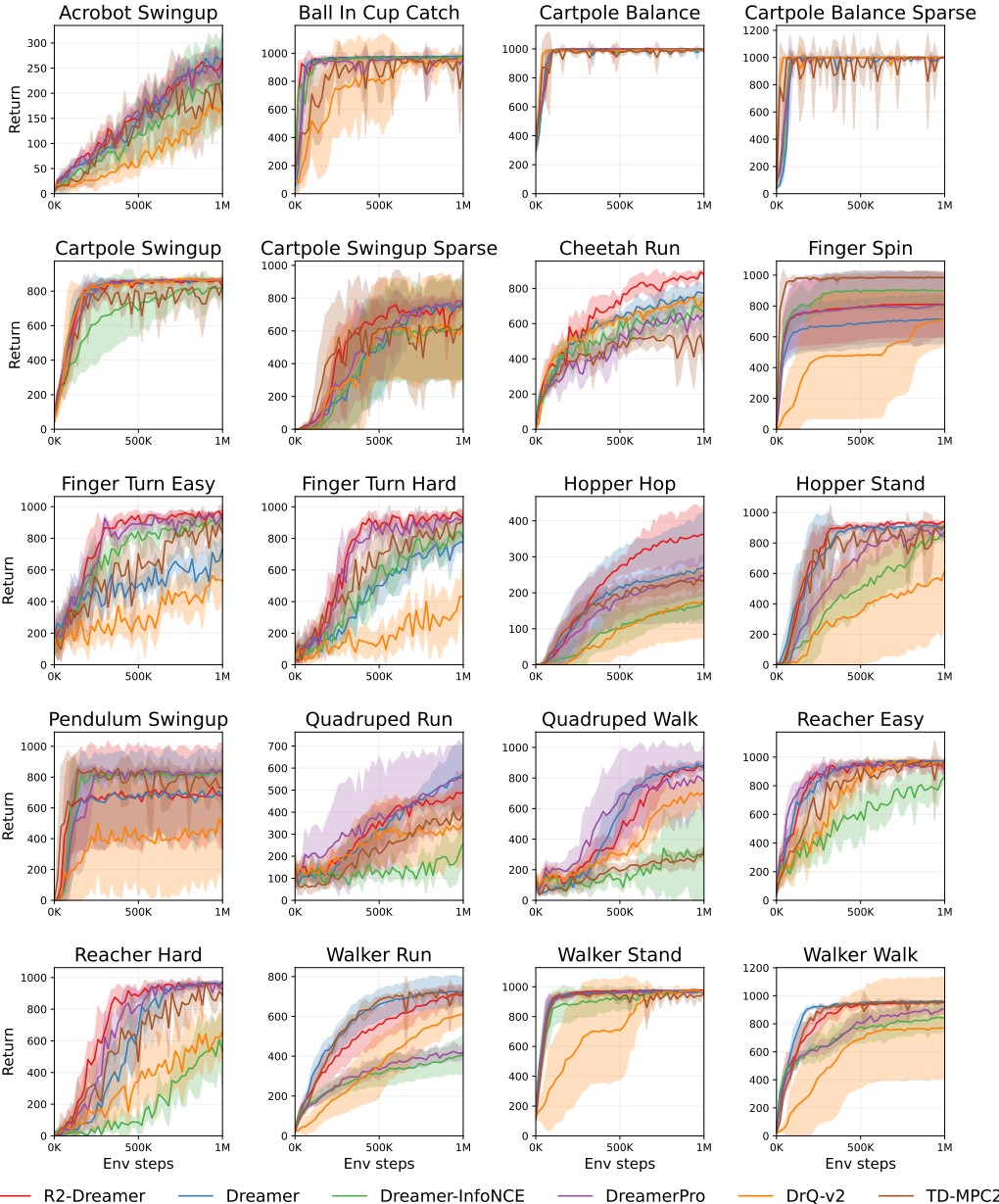

Figure 10: Per-task learning curves for all 20 DMC tasks.

# D DETAILED RESULTS ON META-WORLD

This section provides the individual learning curves for all 50 tasks in Meta-World (Figure 11). The corresponding aggregated results (mean/median success rate across tasks) are shown in Figure 4 in the main text.

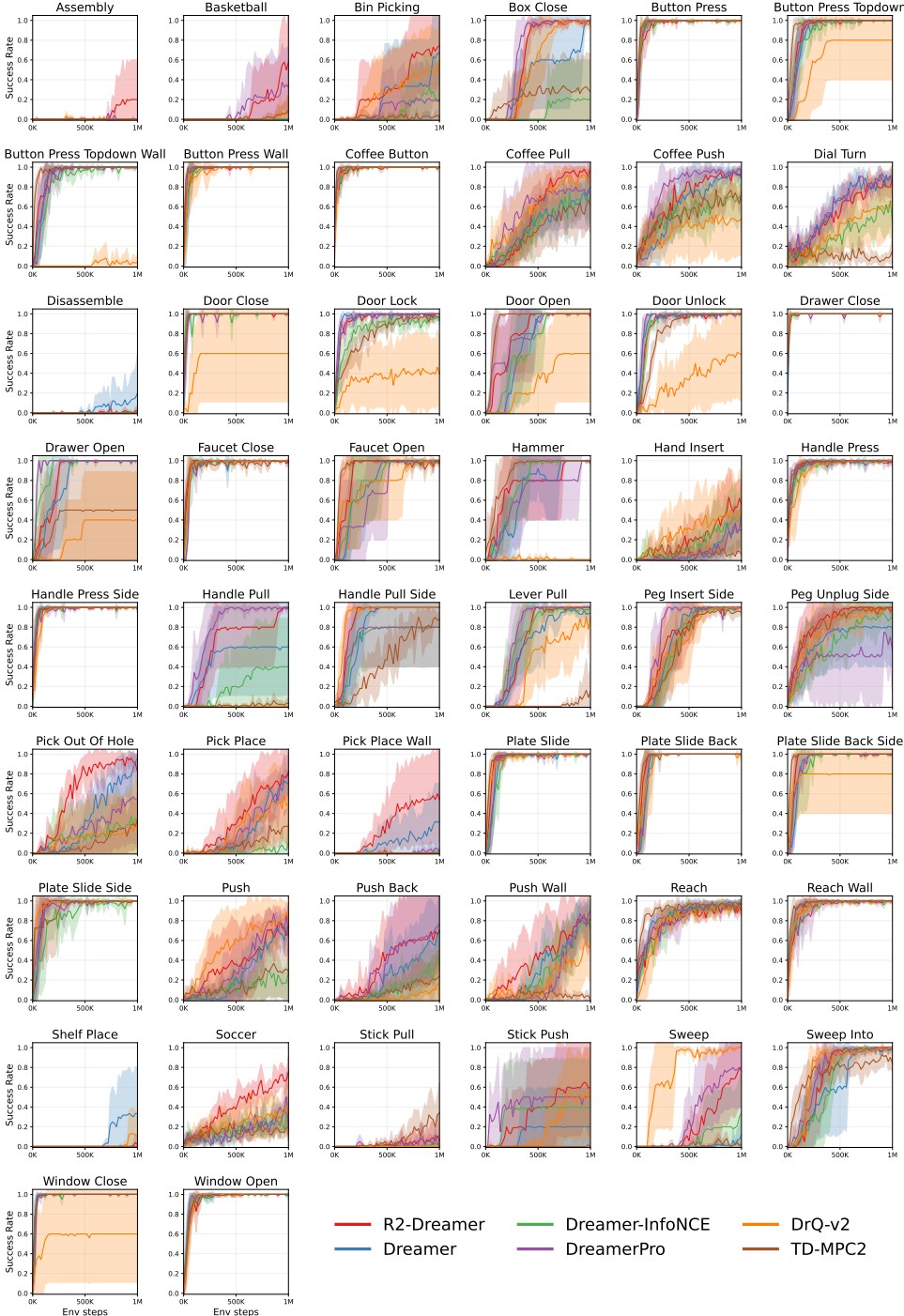

Figure 11: Per-task learning curves for all 50 Meta-World tasks.

# E   DETAILED RESULTS ON ABLATION STUDIES

This section provides the individual learning curves for all 20 tasks in our ablation study (Figure 12). The corresponding aggregated results (mean/median across tasks) are shown in Figure 7 in the main text.

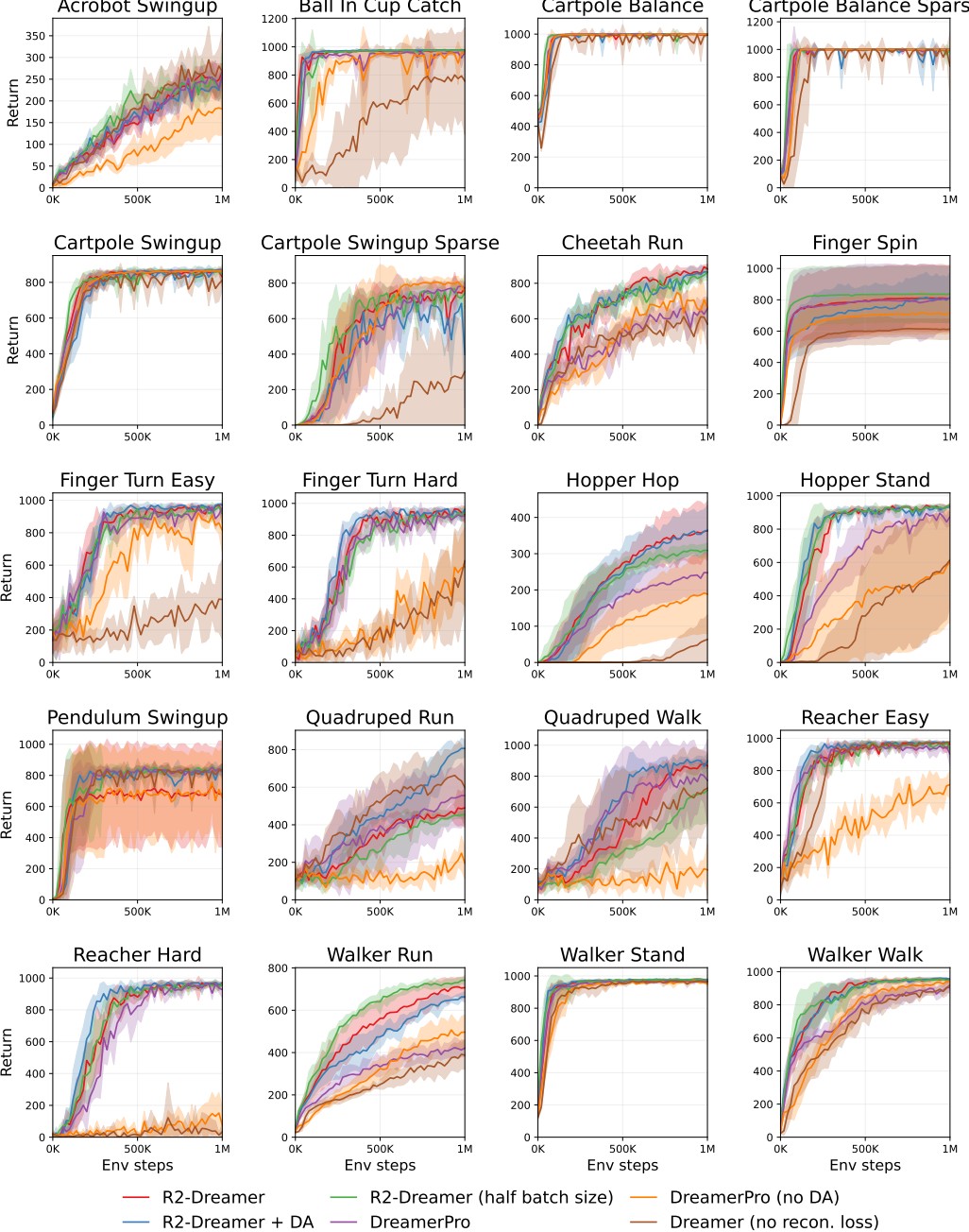

Figure 12: Per-task learning curves for the ablation study across all 20 DMC tasks.

# F  HYPERPARAMETERS

Table 2 summarizes the hyperparameters used in this study. These settings are based on those of DreamerV3, with minimal modifications related to the proposed representation learning objective. We used a single fixed hyperparameter configuration across all benchmarks, without task- or environment-specific adjustments unless otherwise stated.

Table 2: Main hyperparameters. Our settings are identical to DreamerV3, with key changes to the representation learning loss.

| Name | Symbol | Value |
|---|---|---|
| **General** | | |
| Replay capacity | — | $5 \times 10^6$ |
| Batch size | $B$ | 16 |
| Batch length | $T$ | 64 |
| Activation | — | $\mathrm{RMSNorm} + \mathrm{SiLU}$ |
| Learning rate | — | $4 \times 10^{-5}$ |
| Gradient clipping | — | $\mathrm{AGC}(0.3)$ |
| Optimizer | — | $\mathrm{LaProp}(\epsilon = 10^{-20})$ |
| **World Model** | | |
| BT loss scale | $\beta_{\mathrm{BT}}$ | 0.05 |
| Redundancy loss scale | $\alpha$ | $5 \times 10^{-4}$ |
| Dynamics loss scale | $\beta_{\mathrm{dyn}}$ | 1 |
| Representation loss scale | $\beta_{\mathrm{rep}}$ | 0.1 |
| Latent unimix | — | 1% |
| Free nats | — | 1 |
| **Actor-Critic** | | |
| Imagination horizon | $H$ | 15 |
| Discount horizon | $1/(1-\gamma)$ | 333 |
| Return lambda | $\lambda$ | 0.95 |
| Critic loss scale | $\beta_{\mathrm{val}}$ | 1 |
| Critic replay loss scale | $\beta_{\mathrm{repval}}$ | 0.3 |
| Critic EMA regularizer | — | 1 |
| Critic EMA decay | — | 0.98 |
| Actor loss scale | $\beta_{\mathrm{pol}}$ | 1 |
| Actor entropy regularizer | $\eta$ | $3 \times 10^{-4}$ |
| Actor unimix | — | 1% |
| Actor RetNorm scale | $S$ | $\mathrm{Per}(R, 95) - \mathrm{Per}(R, 5)$ |
| Actor RetNorm limit | $L$ | 1 |
| Actor RetNorm decay | — | 0.99 |

# G  PSEUDOCODE FOR REPRESENTATION LOSS

Algorithm 1 provides a PyTorch-style pseudocode for the core representation learning objective.

---

**Algorithm 1** R2-Dreamer Representation Loss (PyTorch-style Pseudocode)

---

```python
# alpha: weight on the off-diagonal terms
# B: batch size, T: time steps, D: feature dimension
# h: deterministic state from sequence model, [B, T, H_dim]
# z: stochastic state from representation model, [B, T, Z_dim]
# e: embeddings from image encoder, [B, T, E_dim]
# projector: linear layer to project concatenated state to embedding
    space

# Project features from dynamics model
state = torch.cat([h, z], dim=-1)
k = projector(state)  # [B, T, D]

# Reshape for loss computation
k = k.reshape(B * T, D)
e = e.detach().reshape(B * T, D)  # Stop gradient to encoder

# Normalize features along the batch dimension
k_norm = (k - k.mean(dim=0)) / (k.std(dim=0) + 1e-5)
e_norm = (e - e.mean(dim=0)) / (e.std(dim=0) + 1e-5)

# Cross-correlation matrix
C = (k_norm.T @ e_norm) / (B * T)  # [D, D]

# Invariance loss
invariance_loss = ((torch.diagonal(C) - 1)**2).sum()

# Redundancy reduction loss
off_diag = C.clone()
off_diag.fill_diagonal_(0)
redundancy_loss = (off_diag**2).sum()

# Total loss
loss = invariance_loss + alpha * redundancy_loss
```

---

# H  THE USE OF LARGE LANGUAGE MODELS

We utilized large language models to improve the grammar and readability of this manuscript.

