# OpenReview forum: "R2-Dreamer: Redundancy-Reduced World Models without Decoders or Augmentation"
_ICLR.cc/2026/Conference — ICLR 2026 Poster_

### Official Review · Reviewer_zUBH · 2025-10-25

**Soundness:** 2
**Presentation:** 2
**Contribution:** 2
**Rating:** 4
**Confidence:** 4

**Summary:**

This paper  presents a decoder-free agent that introduces a self-supervised objective acting as an internal regularizer to prevent collapse. Experiments on DMC and DMC-Subtle validate the effectiveness of the proposed method.

**Strengths:**

- This paper proposes a decoder-free MBRL agent that adopt internal regularizer to avoid reconstructing observations from a latent state.
- This paper introduces DMC-Subtle, a modified DMC benchmark where task-critical objects’ sizes are significantly reduced, demanding a higher level of representational precision.
- The proposed approach outperforms baselines in DMC and DMC-Subtle.

**Weaknesses:**

- There is no comparison with competitive baselines such as TD-MPC2 [1].
- The proposed DMC-Subtle benchmark seems somewhat questionable or not well-justified.
- More challenging benchmarks are needed to better validate the effectiveness of the proposed algorithm.
- The DMC tasks used are relatively simple. How does the method perform on more complex tasks like *DMC Dog* or *DMC Humanoid*?
- As shown in Figure 3, the improvement appears to be marginal.
- There is no comparison with VAI [2], which is a baseline that adopts unsupervised visual attention.

References:

[1] Hansen et al. "TD-MPC2: Scalable, Robust World Models for Continuous Control", ICLR, 2024.

[2] Wang et al. "Unsupervised Visual Attention and Invariance for Reinforcement Learning", CVPR, 2021.

**Questions:**

Have the authors tried conducting experiments on environments with distractors, such as Distracting Control Suite [1]?

Reference:

[1] Stone et al. "The Distracting Control Suite — A Challenging Benchmark for Reinforcement Learning from Pixels", arXiv, 2021.

---

> ### Author Response · Authors · 2025-11-25
>
> We thank the reviewer for the constructive feedback and for acknowledging the strengths of our approach, including the introduction of the internal regularizer and the DMC-Subtle benchmark. We address your concerns below.
>
> **1. Comparison with TD-MPC2**
>
> > *There is no comparison with competitive baselines such as TD-MPC2...*
>
> We thank the reviewer for suggesting this comparison, as it helps contextualize our method against state-of-the-art decoder-free approaches. We added a comparison with TD-MPC2. As shown in **Figure 3** (DMC), **Figure 4** (Meta-World), and **Figure 5** (DMC-Subtle), R2-Dreamer remains competitive on standard benchmarks and significantly outperforms TD-MPC2 on precision-demanding tasks, highlighting the effectiveness of our objective in capturing fine-grained spatial features.
>
> **2. Justification of DMC-Subtle Benchmark**
>
> > *The proposed DMC-Subtle benchmark seems somewhat questionable or not well-justified.*
>
> DMC-Subtle serves as a **controlled testbed** to isolate and evaluate **spatial precision**, a capability critical for real-world robotics (e.g., inserting keys, picking up small objects) but often overlooked in standard benchmarks where rough localization suffices. By minimizing target sizes, it rigorously tests whether an algorithm can maintain spatial fidelity without being distracted. Crucially, the strong performance of R2-Dreamer on Meta-World—which involves complex manipulation of small objects (e.g., "pick out of hole", "soccer")—empirically validates the relevance of this benchmark. It confirms that the spatial precision isolated by DMC-Subtle is indeed a prerequisite for success in more complex, contact-rich manipulation tasks.
>
> **3. More Challenging Benchmarks (Meta-World)**
>
> > *More challenging benchmarks are needed...*
>
> We appreciate the reviewer's suggestion to evaluate on more challenging benchmarks, which has strengthened our empirical evaluation. We added evaluation on **Section 4.3 (Performance on Meta-World)**. While DMC serves as a standard benchmark for locomotion and basic control, Meta-World poses a complementary challenge by focusing on diverse robotic manipulation tasks that involve complex object interactions and fine-grained contact dynamics. R2-Dreamer demonstrates competitive performance in these tasks, confirming its versatility across diverse task domains.
>
> **4. Task Complexity**
>
> > *The DMC tasks used are relatively simple. How does the method perform on more complex tasks like DMC Dog or DMC Humanoid?*
>
> Our evaluation includes **Quadruped** (action dim 12, relatively complex dynamics), suggesting scalability to higher-dimensional systems. We leave Humanoid for future work due to computational constraints, as it requires significantly longer training times for comparison. We have also added this as a future direction in **Section 5 (Conclusion)**.
>
> **5. Marginal Improvement in Figure 3**
>
> > *As shown in Figure 3, the improvement appears to be marginal.*
>
> While performance on standard tasks is comparable to SOTA (which is our goal given the removal of the decoder), R2-Dreamer achieves this with **1.59x faster training** (**Table 1** in **Section 4.7**), thanks to its simple decoder-free design. **We have updated the Abstract to explicitly highlight this computational efficiency.** Furthermore, the significant gains in **Figure 5** (DMC-Subtle) demonstrate its unique advantage in precision.
>
> **6. Comparison with VAI**
>
> > *There is no comparison with VAI...*
>
> We appreciate the reviewer for highlighting VAI, which is a pioneering work in unsupervised visual attention for RL. We have added it to our **Section 2 (Related Work)**.
> While VAI's motion-based attention is highly effective for dynamic scenes, our evaluation contains tasks where **static** but critical objects (e.g., fixed targets) must be identified in scenarios where motion cues are absent. Given this distinction in problem setting, we believe a direct comparison would not fully reflect VAI's strengths or our method's focus.
>
> **7. Distracting Control Suite**
>
> > *Have the authors tried conducting experiments on environments with distractors...?*
>
> We thank the reviewer for this insightful suggestion, which highlights a critical aspect of robustness. We agree this is a promising direction. We have updated **Section 5 (Conclusion)** to highlight testing on the Distracting Control Suite as key future work, given our method's potential robustness to irrelevant backgrounds.
>
> ---
>
> Your comprehensive feedback has been invaluable. We believe that addressing your points—particularly by adding Meta-World and TD-MPC2 comparisons—has greatly improved the manuscript's quality and completeness. We remain available for any further discussions during the rebuttal period.

---

> > ### Comment · Reviewer_zUBH · 2025-11-26
> >
> > Thanks for your response. I will raise my score.

---

### Official Review · Reviewer_Eb9R · 2025-10-27

**Soundness:** 3
**Presentation:** 4
**Contribution:** 3
**Rating:** 8
**Confidence:** 4

**Summary:**

This paper proposes R2-Dreamer, an enhanced variant of DreamerV3 that introduces an internal regularizer to improve latent state representation learning. Specifically, the authors replace the traditional image reconstruction loss with a Barlow Twins loss, aiming to encourage more informative and less redundant latent features. Theoretical analysis shows that this new objective is equivalent to optimizing a variational bound on an extended Sequential Information Bottleneck. Experimental results demonstrate that the proposed method achieves more robust latent representations and improved computational efficiency.

**Strengths:**

1. The paper clearly identifies a key limitation of DreamerV3: its latent representations can be overly influenced by reconstructing input images, leading to a focus on irrelevant pixels rather than task-relevant features. By replacing the reconstruction loss with a self-supervised objective defined in the latent space, the proposed method encourages more compact and task-relevant representations.
2. The paper provides solid theoretical analysis to explain the impact of the new learning objective and includes extensive experimental validation. Results on the challenging DMC-Subtle benchmark demonstrate the superiority of the proposed representation. Moreover, the ablation studies convincingly show both the necessity of the proposed objective and the redundancy of data augmentation under this framework.
3. The proposed approach is conceptually simple yet empirically effective, making it a strong candidate for a new baseline in model-based reinforcement learning.

**Weaknesses:**

1. The experimental environments are not sufficiently diverse. The paper evaluates only on DMC-Subtle, whereas DreamerV3 has also been tested on other benchmarks such as Atari and DMLab. Including experiments on these additional environments would make the evaluation more comprehensive and the conclusions more convincing.

**Questions:**

1. Minor: Why are the ablation results of R2-Dreamer+DA different between Figure 7 and Figure 10? In Figure 7, the performance of the DA version shows a significant drop, while in Figure 10, the drop is much smaller. This discrepancy seems inconsistent with the claim that data augmentation is harmful for precision-demanding tasks.

---

> ### Author Response · Authors · 2025-11-25
>
> We thank the reviewer for the positive evaluation and for highlighting the conceptual simplicity and empirical effectiveness of our proposed method. We appreciate the detailed feedback regarding experimental diversity and ablation consistency.
>
> **1. Diversity of Experimental Environments**
>
> > *The experimental environments are not sufficiently diverse... Including experiments on these additional environments would make the evaluation more comprehensive...*
>
> We agree that demonstrating performance across diverse environments is crucial. We prioritized the **Meta-World** benchmark to evaluate applicability to complex robotic manipulation, which aligns closely with our focus on precise control.
> As detailed in the new **Section 4.3 (Performance on Meta-World)**, our method achieves competitive performance across these tasks. This confirms that our method's ability to capture spatial features (demonstrated in DMC-Subtle) generalizes effectively to manipulation tasks.
>
> **2. Inconsistency in Ablation Studies**
>
> > *Why are the ablation results of R2-Dreamer+DA different between Figure 7 and Figure 10? ... This discrepancy seems inconsistent with the claim that data augmentation is harmful for precision-demanding tasks.*
>
> This difference actually reinforces our core claim about the risks of heuristic DA.
> *   **Standard DMC (Figure 12 in revised paper):** Targets are large; random shift is benign or helpful.
> *   **DMC-Subtle (Figure 8 in revised paper):** Targets are minute; random shift distorts critical spatial features, causing significant drops.
>
> This demonstrates that while heuristic DA is effective for some tasks, it risks degrading performance in precision-demanding scenarios—a risk R2-Dreamer avoids by design.
>
> ---
>
> We thank you again for identifying the need for broader evaluation. We hope the new Meta-World results and the clarification on the ablation study satisfactorily address your concerns about experimental diversity and consistency. We remain available for any further discussions during the rebuttal period.

---

> > ### Comment · Reviewer_Eb9R · 2025-11-28
> >
> > Thanks for the responses! I will keep my original score and recommend for acceptance.

---

### Official Review · Reviewer_z3Lo · 2025-10-30

**Soundness:** 2
**Presentation:** 3
**Contribution:** 2
**Rating:** 4
**Confidence:** 3

**Summary:**

This paper proposes a data-augmentation-free learning method for RSSM-based decoder-free MBRL. The core of this method is is a feature redundancy reduction objective inspired by Barlow Twins and therefore has no need for pixel-wise reconstruction or data augmentation. Emprical study shows the superior performance on standard DMC benchmark and more chanlleging DMC-Subtle benchmark.

**Strengths:**

1. The paper is clearly written, allowing readers to follow the main arguments.

2. It provides a comprehensive experiments and ablations to demonstrate the effectiveness of their proposed new representation learning paradigm.

3. Releasing codebase is good, which can facilitate future research.

**Weaknesses:**

1. The authors didn't compare TD-MPC2 in their experiments, which is a strong state-of-the-art baseline for decoder-free methods.
2. Though authors evaluated different methods on many tasks on the DMC benchmark, I think evaluating only on these locomotion tasks is kind of not comprehesive and it would be better to evaluate on other types tasks like Meta-World.
3. One of claims in this paper is that their method doesn't need hand-engineered data augmentation like other decoder-free MBRL methods. But I question whether this is really a significant problem. For example, TD-MPC2 only uses very simple random shift augmentation, which I don't think it is troublesome for algo implementation. So I'm confused that why the data augmentation for decoder-free MBRL methods is a drawback.

**Questions:**

1. Could the authors compare their method with state-of-the-art decoder-free MBRL methods like TD-MPC2?
2. Could the authors evaluate different methods on other benchmarks like Meta-World?
3. Could authors explain why data augmentation is a problem?

---

> ### Author Response · Authors · 2025-11-25
>
> We thank the reviewer for the constructive feedback and for acknowledging the clarity of our writing and the comprehensiveness of our ablation studies. We are encouraged that the reviewer sees value in our code release. Below, we address the weaknesses and questions raised.
>
> **1. Comparison with TD-MPC2**
>
> > *The authors didn't compare TD-MPC2 in their experiments...*
>
> Thank you for suggesting this strong baseline. We have added a comparison with the official TD-MPC2 implementation. As shown in the updated figures (e.g., **Figure 3** for DMC, **Figure 4** for Meta-World, and **Figure 5** for DMC-Subtle), R2-Dreamer achieves competitive performance on standard benchmarks and significantly outperforms TD-MPC2 on **DMC-Subtle**. This demonstrates that our redundancy reduction objective is more effective than TD-MPC2's approach at capturing fine-grained, task-critical features without relying on data augmentations.
>
> **2. Evaluation on Meta-World**
>
> > *...it would be better to evaluate on other types tasks like Meta-World.*
>
> We appreciate the suggestion to broaden our evaluation. We have added experiments on the **Meta-World** benchmark. As detailed in the new **Section 4.3 (Performance on Meta-World)** and **Appendix D (Detailed Results on Meta-World)**, R2-Dreamer demonstrates strong performance across these diverse manipulation tasks, confirming its versatility across diverse task domains.
>
> **3. Clarification on "Hand-Engineered" Data Augmentation**
>
> > *One of claims in this paper is that their method doesn't need hand-engineered data augmentation... I question whether this is really a significant problem.*
>
> While random shift is **generally applicable**, our core concern is that it can **distort task-critical spatial information** in precision-demanding scenarios. Our experiments on DMC-Subtle demonstrate that such distortions actively degrade performance, whereas R2-Dreamer avoids this issue via internal regularization. We have revised **Section 1 (Introduction)** and **Section 2 (Related Work)** to clarify that the risk of heuristic DA lies in compromising representation fidelity for precision tasks.
>
>
> ---
>
>
> We thank the reviewer again for these constructive suggestions, which have significantly strengthened our paper. By incorporating TD-MPC2 comparisons and Meta-World experiments, we have demonstrated the generality of our approach more robustly. We hope these revisions satisfactorily address your concerns regarding baselines and task diversity. We remain available for any further discussions during the rebuttal period.

---

> > ### Comment · Reviewer_z3Lo · 2025-11-25
> >
> > Most of my concerns have been addressed, but one issue remains:
> > - In the original TDMPC-2 paper, the method clearly outperforms Dreamer on both DMC and Meta-World. However, in the authors’ experiments, TDMPC-2 performs slightly worse than Dreamer.
> > Could you clarify the reason for this discrepancy?

---

> > > ### Author Response · Authors · 2025-11-25
> > >
> > > Thank you for the prompt response. We believe the discrepancy stems from two main factors:
> > > - **Modality**: The TD-MPC2 paper primarily evaluates on **state observations (except for Figure 10)**, where its reward-centric objective excels. However, it may struggle when required to extract critical information from raw images, resulting in limited performance compared to state-based settings.
> > > - **Baseline Version**: We attribute the slight difference in DreamerV3's performance to its **major update in April 2024 [1]**. The TD-MPC2 paper used the older codebase, whereas our experiments utilize the updated version. We have updated the paper to explicitly state that we use the latest DreamerV3. We hope this clarification addresses your concerns.
> > >
> > > References: [1] Hafner, "DreamerV3 Source Code (commit 2411f7d)", GitHub 2024.
> > >
> > > ---
> > >
> > > We thank you for helping us clarify this discrepancy, and we hope this response addresses your concerns.

---

> > > > ### Comment · Reviewer_z3Lo · 2025-11-26
> > > >
> > > > Thank you for the clarification. I have updated the score accordingly.

---

### Official Review · Reviewer_Hody · 2025-11-01

**Soundness:** 4
**Presentation:** 4
**Contribution:** 4
**Rating:** 8
**Confidence:** 2

**Summary:**

The paper proposes R2-Dreamer, a Model-Based Reinforcement Learning (MBRL) agent based on the DreamerV3 architecture. It addresses two main limitations in current image-based world models: the computational expense and potential for task-irrelevant overfitting in decoder-based models, and the reliance on brittle, task-specific Data Augmentation (DA) in existing decoder-free models. R2-Dreamer replaces the reconstruction objective with a redundancy-reduction objective inspired by Barlow Twins, applied between the image embeddings and the projected latent states of the RSSM.

**Strengths:**

1. Principled approach to removing DA: The paper successfully identifies a major pain point in current decoder-free methods—the reliance on heuristic DA. Proposing an information-theoretic internal regularizer (redundancy reduction) as a replacement is a sound and theoretically motivated direction, nicely grounded in the Sequential Information Bottleneck framework in Appendix A2.
2. Strong empirical validation of the core hypothesis: The ablation study (Figure 6) provides compelling evidence. It shows that while DreamerPro collapses without DA, R2-Dreamer maintains performance, proving that the proposed $\mathcal{L}_{BT}$ effectively prevents collapse without external heuristics
3. Effective stress-testing (DMC-Subtle): DMC-Subtle cleanly isolates failure modes of reconstruction-based (wasted capacity on background) and DA-based (distortion of small features) methods, highlighting the specific advantages of R2-Dreamer in precision-demanding tasks
4. Computational Efficiency: The reported speedups are significant and practically important for scaling MBRL.

**Weaknesses:**

1. Batch size concerns for Barlow Twins: Redundancy reduction objectives often require large batch sizes for stable covariance estimation. The paper uses standard Dreamer batching ($B=16, T=64 \implies N=1024$)8. While this appears sufficient for DMC, it raises concerns about stability in higher-dimensional or more diverse visual environments where 1024 samples might not sufficiently estimate the cross-correlation matrix.
2. Ambiguity in Encoder Training: The pseudocode (Algorithm 1) indicates that the image embeddings $e$ are detached before entering the $\mathcal{L}_{BT}$ loss9. If accurate, this means the image encoder does not receive gradients from the primary representation learning objective, relying solely on gradients flowing back from the RSSM via $KL$ terms, which is highly unusual for this class of SSL objectives.

**Questions:**

1. Clarification on Gradients: In Algorithm 1, you have e = ...detach(). Does this mean your image encoder $f_\phi(x_t)$ receives NO gradients from $\mathcal{L}_{BT}$? If so, what is the primary learning signal for the encoder? Is it solely the $KL(q(z_t|h_t, e_t) || p(z_t|h_t))$ term?
2. Batch Size Sensitivity: Have you evaluated the sensitivity of R2-Dreamer to the batch size (specifically the total samples $B \times T$ used for the correlation matrix)? Barlow Twins often degrades rapidly with small batches.
3. Complex Backgrounds: How does R2-Dreamer perform when the background is not just static (like DMC) but dynamic and irrelevant (e.g., "distracting control" suite)? This would be a stronger test of the claim that it avoids wasting capacity on irrelevant details compared to reconstruction.

---

> ### Author Response · Authors · 2025-11-25
>
> We thank the reviewer for the positive assessment and for recognizing the principled approach of our method, the strong empirical validation, and the effectiveness of our stress-testing with DMC-Subtle. We address your concerns and questions below.
>
> **1. Batch Size Sensitivity**
>
> > *Have you evaluated the sensitivity of R2-Dreamer to the batch size...?*
>
> We appreciate this insightful question. While SSL methods often require large batches, the original Barlow Twins paper [1] demonstrates that it is relatively more robust to smaller batch sizes compared to other methods like BYOL or SimCLR.
> To verify if this robustness holds in our setting, we conducted a sensitivity analysis on the Standard DMC benchmark.
> The results, added to **Figure 7**, show that reducing the batch size by half (from 16 to 8) **does not lead to any significant performance drop**. This stability aligns with the properties of Barlow Twins reported in the original paper. In our sequential setting, we hypothesize that the temporal dimension further aids covariance estimation by increasing the effective sample diversity within each batch.
>
> **2. Clarification on Gradients and Encoder Training**
>
> > *In Algorithm 1, you have e = ...detach(). Does this mean your image encoder receives NO gradients...? What is the primary learning signal for the encoder?*
>
> It is correct that the target encoder embeddings are detached for stability, similar to TD-MPC2 [2]. However, the encoder receives rich gradients flowing back through the RSSM from the Invariance and Redundancy terms, as well as from the reward, continuation, value, and dynamics objectives. We have clarified this gradient flow in the revised **Section 3.2 (World Model Learning)**.
>
> **3. Complex Backgrounds**
>
> > *How does R2-Dreamer perform when the background is not just static... but dynamic and irrelevant...?*
>
> While we have not conducted experiments on the Distracting Control Suite, our results on DMC-Subtle demonstrate that R2-Dreamer effectively avoids wasting capacity on irrelevant static background pixels. We hypothesize this property extends to dynamic backgrounds. We have updated **Section 5 (Conclusion)** to highlight the verification of this hypothesis in environments like the Distracting Control Suite as a key direction for future work.
>
> **References:**
>
> [1] Zbontar et al., "Barlow Twins: Self-Supervised Learning via Redundancy Reduction", ICML 2021.
>
> [2] Hansen et al. "TD-MPC2: Scalable, Robust World Models for Continuous Control", ICLR, 2024.
>
> ---
>
> We hope that our response and the updated manuscript have satisfactorily addressed your concerns. We remain available for any further discussions during the rebuttal period.

---

> > ### Comment · Reviewer_Hody · 2025-11-26
> >
> > Thank you for the responses! I will keep my original score and recommend for acceptance.

---

### Meta-Review · Area_Chair_y87r · 2026-01-08

**Summary:**

This paper proposes a variant of DreamerV3, called R2-Dreamer, that removed decoder and introduces an internal regularizer to improve latent state representation learning. Specifically, the authors replace the traditional image reconstruction loss with a Barlow Twins loss, aiming to encourage more informative and less redundant latent features. Experimental results demonstrate that the proposed method achieves more robust latent representations and improved computational efficiency.

Overall, after the rebuttal, reviewers found most of the concerns have been addressed and all the reviewers reach a consensue of possitve recommendation for this paper.

**Reviewer Concerns:**

One category of concerns pertains to the experimental settings, including sensitivity to batch size, comparisons with TD-MPC2, evaluation on Meta-World, and the overall diversity of the experimental environments. These were effectively addressed through additional experiments provided in the rebuttal. However, concerns regarding a comparison with VAI and testing on more complex tasks, such as DMC Dog or DMC Humanoid, remain outstanding.

A second category of concerns involves clarifications on gradient flow, encoder training, data augmentation, and the justification for the DMC-subtle benchmark. These points have been successfully clarified in the revised manuscript.

**Reviewer Scores:**

The paper originally received scores of 8, 8, 4, and 4. Since the majority of the concerns have been addressed, both reviewers who initially gave a score of 4 have indicated an intention to adjust their scores upward

---

### Decision · Program_Chairs · 2026-01-26

Accept (Poster)